# Evaluation of Retail Meat as a Source of ESBL *Escherichia coli* in Tamaulipas, Mexico

**DOI:** 10.3390/antibiotics11121795

**Published:** 2022-12-10

**Authors:** Ana Verónica Martínez-Vázquez, Antonio Mandujano, Eduardo Cruz-Gonzalez, Abraham Guerrero, Jose Vazquez, Wendy Lizeth Cruz-Pulido, Gildardo Rivera, Virgilio Bocanegra-García

**Affiliations:** 1Centro de Biotecnología Genómica of Instituto Politécnico Nacional, Reynosa 88710, Mexico; 2CONACyT Research, Centro de Investigación en Alimentación y Desarrollo, Mazatlán 82112, Mexico; 3Facultad de Medicina Veterinaria, Universidad Autónoma de Tamaulipas, Cd Victoria 87274, Mexico; 4Escuela de Medicina, Campus Reynosa, Universidad del Valle de México, Reynosa 88760, Mexico

**Keywords:** meat, *Escherichia coli*, ESBL, antibiotic resistance, virulence

## Abstract

In recent decades, the appearance of a group of strains resistant to most β-lactam antibiotics, called extended-spectrum β-lactamases (ESBLs), has greatly impacted the public health sector. The present work aimed to identify the prevalence of ESBL-producing *Escherichia coli* strains in retail meat from northeast Tamaulipas. A total of 228 meat samples were obtained from 76 different stores. A prevalence of *E. coli* ESBL of 6.5% (15/228) was detected. All (15/15) of the ESBL strains were multiresistant. Altogether, 40% (6/15) of the strains showed the presence of class 1 integrons. The isolates identified with *bla*_CTX-M_ (20%) also showed co-resistance with the *tet* (A and B), *str* (A and B), and *sul* (2 and 3) genes. A total of 20% of the strains belonged to the B2 and D phylogroups, which are considered pathogenic groups. None of the ESBL-positive strains contained any of the virulence gene factors tested. The presence of ESBL-producing *E. coli* strains in meat indicates a potential risk to the consumer. Although most of these strains were classified as commensals, they were found to serve as reservoirs of multiresistance to antimicrobials and, therefore, are potential routes of dispersion of this resistance to other bacteria.

## 1. Introduction

In the past, antibiotic-resistant bacteria have predominantly been associated with hospital settings, but in recent decades, they have also been identified in water, soil, and animals [1,2,3]. In meat destined for human consumption, mainly beef, pork, and poultry, high percentages of bacteria, such as resistant and multidrug-resistant (MDR) *Escherichia coli*, have been detected [3,4,5]. Some strains of MDR bacteria produce extended-spectrum β-lactamases (ESBLs), which hydrolyze and cause resistance to various types of β-lactam antibiotics. These include broad-spectrum cephalosporins (cefotaxime, ceftriaxone, ceftazidime) and monobactams (aztreonam) [6,7]. Β-lactams likely have the greatest number of chemical modifications of any antibiotic family. Diversification in genes encoding β-lactamases towards a broader spectrum of enzyme activity has been reported [8]. More than 300 ESBL enzymes are known; those with the greatest representation are TEM, SHV, and CTX-M, with the latter having the greatest distribution in the world [9]. Resistance development in these ESBL strains can occur in commensal strains that coexist with their host without causing harm but with the possibility of transmitting resistance to other commensal or pathogenic bacteria, even from other species. A relationship exists between phylogeny and pathogenesis, and from this, a technique has been developed for the detection of four main phylogenetic groups of *E. coli* called A1, B1, B2, and D [10]. Most of the virulent strains responsible for infections are associated with groups D and B2, while commensal strains generally belong to the phylogroups A and B1 [10,11]. In several countries, the presence and virulence of *E. coli* ESBL strains in food of animal origin have been studied, mainly in beef, pork, and chicken. Most of these studies, particularly those in Europe and Asia, obtained variable results from one region to another with a prevalence between 1.1 and 82% [4,6,12,13,14]. In Latin America, scarce data have been published regarding the prevalence of ESBL strains in meat available to the public. In Mexico, the first results published showed a prevalence of *E. coli* ESBL in beef and chicken meat of 35% in the city of Puebla, which is located in the center of the country [15]. However, it is unknown if this prevalence is homogeneous or variable in the rest of Mexico. Thus, the objective of this work was to identify the prevalence and virulence of *E. coli* ESBL strains in meat commercialized in northeast Mexico (Tamaulipas) to estimate the risk that this represents for the consumer.

## 2. Results

### 2.1. Prevalence of E. coli ESBL in Samples

A total of 228 retail meat samples, including pork (*n* = 76), beef (*n* = 76), and chicken (*n* = 76), were purchased from supermarkets and retail stores (butcheries) located in northeast Tamaulipas. One hundred and fifty-one samples were confirmed as *E. coli* based on morphological and biochemical tests showing an overall prevalence of 66.2% (151/228) (Table 1); specifically, the prevalence was 65% in Reynosa (78/120), 66% in Río Bravo (22/33), and 68% in Matamoros (51/75). Of these 228 samples, 64.6% of the *E. coli* isolates presented susceptibility to cefotaxime, 13.1% intermediate resistance, and 22.1% were cefotaxime-resistant. A positive synergy test and combined disc test were observed in isolates from 15 different samples (6.5% ESBL; 15/228). Among the meat samples, the ESBL-producing *E. coli* isolates were more frequently detected in the beef samples, with a prevalence of 40% (6/15), than in the chicken samples, with a prevalence of 33.3% (5/15), or the pork samples at 26.6% (4/15).

### 2.2. Antimicrobial Susceptibility Phenotypes and Genotypes in E. coli ESBL

All 15 ESBL-producing *E. coli* were multidrug-resistant (MDR), demonstrating resistance to at least four antibiotics. In the ESBL-producing strains, resistance to cephalothin (CF), amoxicillin–clavulanic acid (AMC), and cefepime (FEP), with 100% (15/15) prevalence, was most commonly observed, followed by cefotaxime (CTX) with 86.6% (13/15), ceftazidime (CAZ) and ampicillin (AM) with 73.3% (11/15) each, and tetracycline (TE) with 60% (9/15). Resistance to levofloxacin (LEV) and ciprofloxacin (CIP) was observed less frequently, with a prevalence of 13.3% (2/15).

The class 1 integron was detected in 53.3% (8/15) of the ESBL-producing *E. coli* strains; the class 2 and 3 integrons were not identified in these samples (Table 2 and Table 3).

### 2.3. Distribution of Phylogenetic Groups in E. coli ESBL

Most (11) of the ESBL-producing *E. coli* strains were classified into phylogenetic group A, with one being classified into group B1, two into group B2, and one into group D. The ESBL-producing *E. coli* strains were negative for *stx*1, *stx*2, *hly*A, *bfp*, and *eae* according to PCR.

## 3. Discussion

To our knowledge, this is the first study on extended-spectrum β-lactamase *E. coli* strains in commercial meat in northeast Mexico (Tamaulipas state). In the samples studied, the prevalence of *E. coli* was 66.2% (151/228), which suggests the poor hygienic handling of meat. This value is greater than the prevalence of *E. coli* of 55.7% reported in meat from Tamaulipas [16]. The ESBL-producing *E. coli* strains had a prevalence of 6.5% (15/228), with these values being lower than those reported in two other studies on ESBL in Mexico. In 2018, Barrios et al. reported an ESBL-producing *E. coli* prevalence of 35% in pork and chicken [15]. In Puebla, the chicken had a 39% (14/36) ESBL prevalence, while in northeast Tamaulipas, it was 7.8% (6/76). A similar case was observed in the same study with pork; in Puebla, a 20% (2/10) ESBL prevalence was reported, and in this sample, it was 5.2% (4/76). Another study was conducted in Oaxaca, Mexico, by Galindo [17], focusing on beef and pork. It is important to point out that Galindo reported results for enterobacteria as a group and not only *E. coli* strains. In their study, beef had a prevalence of ESBL of 68.8% (22/32), which contrasts with the 6.5% prevalence obtained in our results. For chicken meat, in Oaxaca, the prevalence was 66.7% (12/18) [17], and in this work, it was 7.8% (6/76). In similar studies of meat from different countries, the prevalence of ESBL tended to be greater than that found in this study. Examples include studies published by Ye et al. [18] in China with a 14.5% prevalence (chicken, beef, and pork, with no other type of sample), Randall et al. [13] in the United Kingdom with a 27.5% prevalence (beef, pork, and chicken, not including vegetables), Nguyen et al. [19] in Vietnam with a 45.5% prevalence (chicken, beef, pork, and seafood), and Yu et al. [14] in China with a 16.7% prevalence (chicken, beef, pork, and lamb). Although the presence of ESBL strains in commercial meat from northeast Tamaulipas may suggest a public health risk, its low percentage could indicate that the level of risk is also low. However, 100% (15/15) of these ESBL strains were resistant to 5 to 14 antibiotics, with a greater percentage of resistance to β-lactams and tetracyclines, which are antibiotics that are frequently used in cattle breeding and in which resistant strains have been reported. These strains act as reservoirs of multidrug-resistant genes that can be transferred to strains of the same species or other species and therefore can be a source of antimicrobial resistance [15]. Some of this multiresistance is codified in the gene cassette of class 1, 2, and 3 integrons. In this study, class 1 integrons (*intI* 1) were identified in 40% of the ESBL strains (5/15). However, the presence of class 2 and 3 integrons was not identified, a finding that coincides with the results of Kim et al. [12], who also did not observe integrons. This absence of class 2 integrons can be explained since although class 2 integrons have been reported in pathogenic *E. coli* strains and environmental commensals, their frequency is low since they harbor an inactive class 2 integrase [20]. In the case of class 3 integrons, there are reports of their association with class 1 and 2 in isolates of multiresistant strains [21]. However, it is not surprising that class 3 integrons were not detected in this work since only a few reports describe their occurrence [20]. The ESBL gene most commonly identified in the analyzed isolates of *E. coli* was *bla*_CTX-M_ (20%), as was seen in food-producing animals in Europe [22]. CTX-M lactamase has been reported as the dominant ESBL type and the most common globally [18,23,24]. This could be because CTX-M is found mainly in plasmids, which allows horizontal transfer between *Enterobacteriaceae* and explains the current global dispersion of this enzyme epidemic [22,25]. The location of the ESBL genes in mobile gene elements is what drives their association with other resistance genes, conferring resistance on the antibiotics used in animals and humans, which plays an important role in the co-selection of these ESBL genes [26,27]. In this work, the ESBL strains with the gene *bla*_CTX-M_ exhibited multiresistance with tetracyclines presenting both genes *tet* A and B, streptomycins with the genes *str* A and B, and sulfonamides with the genes *sul* 2 and *sul* 3.

When analyzing the distribution of ESBL-producing E. coli strains in phylogroups, it was seen that most are located in commensal groups A (73.3%) and B1 (6.6%), while only 20% of the strains were found to belong to the pathogenic groups (B2 and D). When these results are compared with those reported by Barrios et al. [15] in Mexico, it can be determined that the majority of strains are also in commensal groups (81.2%). An interesting detail is that the proportion for both commensal groups (A and B) is inverted in the two sampling zones. Thus, while in Tamaulipas the current study presented 73% in group A and 6.6% in group B, in Puebla, in the work by Barrios et al., the situation was reversed, with 6.2% in group A and 75% in group B. The high prevalence of commensal *E. coli* ESBL strains indicates a fundamental and silent role carried out by these commensal isolates in the dissemination of ESBL resistance genes [28].

On the other hand, in the results of this study, in the 20% of strains identified for groups B2 and D (considered pathogenic), none of the virulence genes included in this analysis (*stx*1, *stx*2, *hly*A, *eae*, and *bfp*) were identified.

## 4. Materials and Methods

### 4.1. Sample Collection

From September 2017 to September 2018, raw meat samples were collected from supermarkets and retail stores (butcheries) located in 3 cities in northeast Tamaulipas, Mexico: Reynosa, Rio Bravo, and Matamoros. The raw meat samples included ground beef, ground pork, and chicken legs; all were purchased randomly, in 500 g packages. The samples were kept in ice containers and transported to the laboratory.

### 4.2. Isolation and Identification of E. coli

Primary strain isolation was conducted according to the national Mexican standard for pathogen detection in foods (NOM-210-SSA-2014). Briefly, 25 g of each sample was homogenized with 225 mL of lactose broth after incubation at 37 °C for 18–24 h. Following incubation, the samples were cultured on Eosin Methylene Blue (EMB) agar media and MacConkey (MAC) agar (BD Becton Dickinson and Co. NJ, USA) plates and incubated at 37 °C for 18–24 h. Presumptive *E. coli* colonies were identified by standard biochemical tests, including glucose fermentation, indole production, the methyl red test, the Voges–Proskauer test, and citrate utilization.

### 4.3. Phenotypic Test for the Presence of Extended-Spectrum β-Lactamase (ESBL)

The *E. coli* isolates were tested for antimicrobial susceptibility to cefotaxime (CTX 30 μg) using the Kirby–Bauer disc diffusion method according to the guidelines of the Clinical and Laboratory Standards Institute [29]. Muller–Hinton agar plates were prepared with an inoculum standardized to 0.5 McFarland. After placing the antimicrobial disc on the inoculated plates, the plates were incubated at 37 °C for 18–24 h. All the isolates were classified as sensitive, intermediate, or resistant per the recommendations of the CLSI [29]. The resistant and intermediate cefotaxime strains were screened for ESBL by the double-disc synergy test [30]. The bacterial suspension was prepared according to the 0.5 McFarland turbidity standard and was spread evenly on Mueller–Hinton agar. A disc of amoxicillin–clavulanic acid (AMC; 20/10 μg) was placed in the middle of the inoculated plate. Discs of ceftazidime (CAZ, 30 μg), cefepime (FEP; 30 μg), and cefotaxime (CTX; 30 μg) were then placed at a 20 mm distance and incubated at 37 °C for 18–24 h.

The production of ESBL was confirmed using the combined disc method according to CLSI methods [29] using discs containing cefotaxime (CTX; 30 μg) and ceftazidime (CAZ; 30 μg) with or without the addition of clavulanic acid (CA; 10 μg). An increase in the inhibition zone of ≥5 mm around the disks of cefotaxime or ceftazidime in combination with clavulanic acid confirmed the ESBL phenotype.

### 4.4. Antimicrobial Susceptibility Testing

Antimicrobial susceptibility testing of phenotypically confirmed ESBL-producing *E. coli* strains was performed using the standard disc diffusion method against a panel of the following antimicrobial agents: tetracycline (TE 30 μg), chloramphenicol (CL 30 μg), trimethoprim–sulfamethoxazole (SXT 25 μg), streptomycin (STR 10 μg), ciprofloxacin (CIP 5 μg), levofloxacin (LEV 5 μg), and ampicillin (AM 10 μg). The isolates were categorized as sensitive, intermediately resistant, or resistant based on the results interpreted following the recommendations of the CLSI [29].

### 4.5. Detection of Integron Sequences

DNA was extracted from bacterial isolates confirmed for ESBL production according to a standard heat lysis protocol and was analyzed for the presence of *intI*1, *intI*2, and *intI*3 genes (encoding in class1, class2, and class3) by polymerase chain reaction (PCR) (Table 2). The PCR mixture contained buffer 1X, MgCl_2_ 25 mM, dNTPs 10 mM, primer 10 mM, and Taq DNA polymerase 5 U. The PCR amplification conditions were as follows: initial denaturation at 95 °C for 1 min, followed by 30 cycles of denaturation at 95 °C for 45 s, annealing at 60 °C for 45 s, extension at 72 °C for 45 s, and a final cycle of amplification at 72 °C for 7 min. The verification of the PCR products was performed using 2.5% agarose gel with 0.5X TBE and at 100 V for 45 min, with a molecular marker (100 pb Promega, WI, USA). The negative control consisted of all contents of the reaction mixture excluding template DNA, which was substituted with 1 μL of nuclease-free water. The DNA bands were visualized and photographed under UV light.

### 4.6. Identification of β-Lactamase Genes

The ESBL-producing isolates were analyzed for the presence of the major beta-lactamase genes *bla*_CTX-M_, *bla*_TEM_, *bla*_SHV_, and *bla*_OXA_ by PCR (Table 2).

### 4.7. Detection of Antibiotic Resistance Genes

Strains confirmed as ESBL were further analyzed for the presence of genes associated with resistance to tetracycline (*tet*A and *tet*B), streptomycin (*str*A and *str*B), gentamicin (*aad*A1 and *aac*3), sulfonamides (*sul*1, *sul*2, and *sul*3), and quinolones (*qnr*A and *qnr*B) and were screened by polymerase chain reaction (PCR) (Table 2).

### 4.8. Phylogenetic Groups

ESBL-producing isolates were classified based on the description described by Clermont et al. [31] into phylogenetic groups (A, B1, B2, and D) according to the presence of the two virulence genes (*chu*A, encoding a hem transporter protein in *E. coli* O157: H7, and *yja*A, initially identified in the genome of *E. coli* K-12) and one DNA fragment, TspE4.C2, by PCR (Table 1) based on Clermont’s method [10,31].

### 4.9. Virulence Factor Genes

ESBL isolates were screened by conventional PCR for five virulence markers: the *stx* genes (*stx*1 and *stx*2), enterohemolysin (*hly*A), bundle-forming pilus (*bfp*), and chromosomal gene (*eae)* using primers and conditions described previously by Canizales et al. [32] (Table 2).

## 5. Conclusions

This is the first study from Tamaulipas showing the presence of multidrug resistance commensal *E. coli* strains with low rates of ESBLs in *E. coli* isolated from retail meat; however, those ESBL-harboring strains may be reservoirs for multidrug resistance genes.

## Figures and Tables

**Table 1 antibiotics-11-01795-t001:** Prevalence of *E. coli* strains in meat samples from northeast Mexico.

City	No. Stores	No. Samples	*E. coli* Prevalence % (*n*)			Total
			Beef	Pork	Chicken	
Reynosa	40	120	57.5% (23/40)	62.5% (25/40)	75.0% (30/40)	65.0% (78/120)
Río Bravo	11	33	72.7% (8/11)	54.5% (6/11)	72.7% (8/11)	66.0% (22/33)
Matamoros	25	75	68.0% (17/25)	76.0% (19/25)	60.0% (15/25)	68.0% (51/75)
Total	76	228	63.1% (48/76)	62.7% (50/76)	69.7% (53/76)	66.2% (151/228)

**Table 2 antibiotics-11-01795-t002:** Antimicrobial resistance genes to ESBL-producing *E. coli* strains according to the type of meat.

Gene	Beef n (%)	Chicken n (%)	Pork n (%)	Total
*bla* _CTX_ _-M_	2/5 (40)	1/6 (17)	0/4 (25)	3/15 (20.0)
*bla* _TEM_	1/5 (20)	1/6 (17)	0/4 (25)	2/15 (13.3)
*bla* _SHV_	0/5 (20)	0/6 (17)	0/4 (25)	0/15 (0.0)
*bla* _OXA_	0/5 (20)	0/6 (17)	0/4 (25)	0/15 (0.0)
*tet*A	2/5 (40)	1/6 (17)	2/4 (50)	5/15 (33.3)
*tet*B	2/5 (40)	3/6 (50)	3/4 (75)	8/15 (53.3)
*str*A	2/5 (40)	1/6 (17)	1/4 (25)	4/15 (26.6)
*str*B	2/5 (40)	1/6 (17)	1/4 (25)	4/15 (26.6)
*sul*1	0/5 (20)	0/6 (17)	1/4 (25)	1/15 (6.6)
*sul*2	2/5 (40)	1/6 (17)	0/4 (25)	3/15 (20.0)
*sul*3	2/5 (40)	1/6 (17)	0/4 (25)	3/15 (20.0)
*qnr*A	0/5 (20)	0/6 (17)	1/4 (25)	1/15 (6.6)
*qnr*B	1/5 (20)	1/6 (17)	1/4 (25)	3/15 (20.0)
*aad*A	1/5 (20)	1/6 (17)	0/4 (25)	1/15 (6.6)
*aac(3)-VI*	0/5 (20)	0/6 (17)	0/4 (25)	0/15 (0.0)
*int1*	4/5 (80)	1/6 (17)	3/4 (75)	8/15 (53.3)
*int2*	0/5 (20)	0/6 (17)	0/4 (25)	0/15 (0.0)
*int3*	0/5 (20)	0/6 (17)	0/4 (25)	0/15 (0.0)

**Table 3 antibiotics-11-01795-t003:** Distribution of resistance genes in *Escherichia coli* isolates from meat.

Tetracycline Resistance				β-Lactamase Resistance			
Phenotypic		Genotypic		Phenotypic		Genotypic	
Tetracycline	9/15 (60.0%)	*tet*A	5/15 (33.3%)	cefotaxime	13/15 (86.6%)	*bla* _CTX-M_	3/15 (20.0%)
		*tet*B	8/15 (53.3%)	amoxicillin–clavulanic acid	15/15 (100%)	*bla* _TEM_	2/15 (13.3%)
		*A* + *B*	4/15 (26.6%)	cefepime	15/15 (100%)	*bla* _SHV_	0
				ceftazidime	11/15 (73.3%)	*bla* _OXA_	0
				ampicillin	11/15 (73.3%)	CTX + TEM	2/15 (13.3%)
Streptomycin resistance				Quinolone resistance			
Phenotypic		Genotypic		Phenotypic		Genotypic	
streptomycin	8/15 (53.3%)	*str*A	4/15 (26.6%)	ciprofloxacin	2/15 (13.3%)	*qnr*A	1/15 (6.6%)
		*str*B	4/15 (26.6%)	levofloxacin	2/15 (13.3%)	*qnr*B	3/15 (20.0%)
		*A* + *B*	4/15 (26.6%)			*A*+*B*	1/15 (6.6%)
		*aad*A	1/15 (6.6%)				
		*A* + *B* + *a*	1/15 (6.6%)				

## Data Availability

Data available upon request.

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
