# Peer review of "Evaluation of Retail Meat as a Source of ESBL Escherichia coli in Tamaulipas, Mexico"

_antibiotics, 2022, doi:10.3390/antibiotics11121795_

Round 1

Reviewer 1 Report

This manuscript reports on an survey of ESBL E. coli on meat in the region of Tamaulipas, Mexico. The data are useful, but not spectacular. I have two suggestions: 1) The text of lines 71 - 85 is hard to digest because of all the number. I suggest to summarize these in a table. 2) In the discussion lines 93-115 are a recap of the introduction and the results. The interesting part starts at line 115.  Either omit of strongly compress this first part. 

Author Response

English language and style are fine/minor spell check required. We corrected the English language and style.

We modified  some sections of the results to make them clear

Suggestion 1: We Modified the redaction, and the results were integrated into table 2.

Suggestion 2: The first lines of the discussion have been removed to compress that part.

Reviewer 2 Report

This study investigated the prevalence of E. coli ESBL strains in meat in northeast Mexico. The prevalence of E. coli ESBL strains was 6.5% (15/228) and all the 15 ESBL strains were multi-drug resistant.

1.    The phylogenetic group analysis is non-specific to show pathogenic potential. And very previous protocol adopted in this study.

2.    Why only test stx1, stx2, hlyA, eae and bfp genes? Other genes should be included to define diarrheagenic E. coli.

3.    What is the relation between antimicrobial phenotype and antimicrobial genes?

Author Response

1. The phylogenetic group analysis is non-specific to show pathogenic potential. And very previous protocol adopted in this study. AUTHORS RESPONSE: The Clermont quadruplex PCR method was used to identify phylogroups since it has proven to be an efficient method, still currently used in recent publications of the last few years, including 2022.

2. Why only test stx1, stx2, hlyA, eae and bfp genes? Other genes should be included to define diarrheagenic E. coli. AUTHORS RESPONSE: Although there are more virulence factors, previous articles from several countries have focused on this set of genes to study Enteropathogenic E. coli (EPEC) and especially EHEC due to its high pathogenicity, based on the method of Canizalez-Roman et al. ., 2013 and/or Malik et al., 2017.

3. What is the relation between antimicrobial phenotype and antimicrobial genes? AUTHORS RESPONSE: A new table 4 was included, where the relationship between phenotypic and genotypic resistance is observed.

The paper was edited for corrections required for the English language and style.

Round 2

Reviewer 2 Report

In the abstract, the authors tried to explore the pathogenesis of ESBL-E. coli isolates by phylogenetic analysis. They also tested the virulence factor genes, but the results were not shown.

 L79-80, “The class 1 integron was detected in 53.3% (8/15) of the ESBL-producing E. coli strains; class 2 and 3 integrons were not identified in these samples (Table 2).” The results did not find in the table.

Author Response

Author's response to reviewer 2 suggestions:

Suggestion 1. In the abstract, the authors tried to explore the pathogenesis of ESBL-E. coli isolates by phylogenetic analysis. They also tested the virulence factor genes, but the results were not shown.

Author's Response: in lines 19-20 in the abstract virulence genes results were added.

Suggestion 2. L79-80, “The class 1 integron was detected in 53.3% (8/15) of the ESBL-producing E. coli strains; class 2 and 3 integrons were not identified in these samples (Table 2).” The results did not find in the table.

Author´s Response: Integron PCR results were added at the end of table 2.